# Strain Modulation of Microstructure, Magnetic Domains, and Magnetic Properties of Ti/Fe/Ni$_{81}$Fe$_{19}$/Fe/Ti Multilayer Thin Films

Zongsheng He [1], Zenan Ma [1], Ziyu Li [1], Yangzhong Du [2], Jun Yang [3], Chuanjian Wu [1] , Qifan Li [1] ,
Xiaona Jiang [1], Chaoming Wang [1], Zhong Yu [1], Zhongwen Lan [1] and Ke Sun [1,*]

1   School of Materials and Energy, University of Electronic Science and Technology of China,
    Chengdu 610054, China
2   Hengdian Group DMEGC Magnetics Company, Ltd., Jinhua 322118, China
3   Guangdong EXSENSE Electronics Technology Company, Ltd., Zhaoqing 526000, China
*   Correspondence: ksun@uestc.edu.cn

**Abstract:** A simple and convenient method is demonstrated in this work by continuously applying uniaxial tensile strains to tune the high-frequency properties of flexible magnetic films. The magnetostriction effect causes the uniaxial magnetic anisotropy in the Ti/Fe/Ni$_{81}$Fe$_{19}$/Fe/Ti multilayer film when the flexible substrate transitions from the convex state to the planar state after preparation. In addition, the microstructure, magnetic domain morphology, and the high-frequency magnetic performance of the pre-strained Ti/Fe/Ni$_{81}$Fe$_{19}$/Fe/Ti multilayer films are investigated. The results show that the flexible Ti/Fe/Ni$_{81}$Fe$_{19}$/Fe/Ti multilayer films' initial permeability can be monotonically varied over a hundred units, and the resonant frequency can be adjusted around 1.5 GHz. The flexible Ti/Fe/Ni$_{81}$Fe$_{19}$/Fe/Ti films, with their elastic-tunable magnetic performance, are promising candidate materials for flexible microwave devices.

**Keywords:** flexible magnetic films; high-frequency properties; magnetic domains; tunable resonance frequency

## 1. Introduction

Microwave magnetic devices are indispensable components in satellite and mobile communications systems [1–6]. The demand for microwave devices such as higher integration, higher operating frequency, and smaller size is an important reflection of the rapid development of the electronic communication industry [7–9]. There is a significant demand to improve the performance of magnetoelectronic units, which are the core components of microwave devices. Further study of the relationship between the magnetic domain structure and the microscopic results of soft magnetic films in the origin of magnetic behavior is an important prerequisite for optimizing magnetic properties [10–13]. In previous studies, most of the magnetic films used in microwave devices were prepared using traditional rigid substrates such as silicon and glass [14–16]. High-frequency microwave magnetic devices are applied to reconfigurable electromagnetic interference shielding and wearable wireless transmission systems, and have been used in the development of flexible substrates. These are the embodiment of the popularity of flexible electronic technology in recent years [17,18]. The flexible high-frequency devices usually show good mechanical strain adjustability [19–22]. Recently, Tang et al. reported amorphous CoFeB films deposited on flexible polyethylene terephthalate (PET) substrates and the magneto-mechanical coupling effect was studied [23]. Liu et al. fabricated flexible CoFeB films grown on pre-strained polydimethylsiloxane, and the high-frequency characteristics of the film were adjusted by changing the applied pre-strain [24]. Flexible magnetic thin films and spintronic devices grown on plastic substrates are conformable, attracting extensive attention. However, the

external strain caused by mechanical deformation can significantly change the magnetic anisotropy due to the magnetostriction effect for magnetic thin films deposited on flexible substrates, [25]. Previous works have shown that strain deposition is one of the simplest and most effective methods to adjust the resonance frequency ($f_r$) of magnetic films [25]. Thin films deposited on flexible substrates are useful in stretchable magnetoelectronics and high-frequency electromagnetic devices due to their strain sensitivity. For practical applications, such as microwave filters and inductors, the $f_r$ is related to strain-induced magnetic anisotropy [26]. In addition, the ability to generate adjustable anisotropy through pre-strain from the flexible substrate bending can eliminate the complex in situ annealing process and clamping effect of the rigid substrate. Therefore, the magnetostrictive thin films deposited on flexible substrates are suitable for studying the relationship between magnetism and external strain. The high-frequency characteristics of magnetic thin films' can be easily tuned by pre-stretching strain.

In this study, we prepared flexible soft magnetic films on PET using a self-made bending die and pre-stretching elastic substrate. The permalloys for changing the anisotropic field $H_k$ is preferred. Fe and $Ni_{81}Fe_{19}$ thin films were deposited on pre-stretched elastomeric substrates by electron beam evaporation. The evolution of magnetic anisotropy with microstructure was studied. It is important to establish the relationship between magnetic anisotropy, texture formation, magnetization reversal, and external strains. The continuous tunability of magnetic films' $f_r$ may pave the way for creating devices with mechanically adjustable resonance frequencies suitable for various frequency-based applications.

## 2. Experimental Details

The TFNFT magnetic film consists of $Ti/Fe/Ni_{81}Fe_{19}/Fe/Ti$ films having a thickness of 2 nm /4 nm/100 nm/4 nm/2 nm, respectively, deposited on the PET flexible substrates using the electron beam evaporation (EB-500). The magnetic anisotropy was controlled by regulating the radius of curvature during the deposition. Figure 1 shows a schematic diagram of the strain induced by bending the flexible PET substrate. Six bases with different curvature were prepared to produce different compressive and tensile strains ($\varepsilon = -0.26\%$, $\varepsilon = -0.52\%$, $\varepsilon = -0.78\%$, $\varepsilon = 0.26\%$, $\varepsilon = 0.52\%$, $\varepsilon = 0.78\%$). Then, different flexible substrates were pasted on the convex mold and the concave mold so that different bending stresses were introduced into the film sample through the flexible substrate. The induced stress on the TFNFT films can be calculated by Hooke's law, which describes the strain–stress relationship in the film.

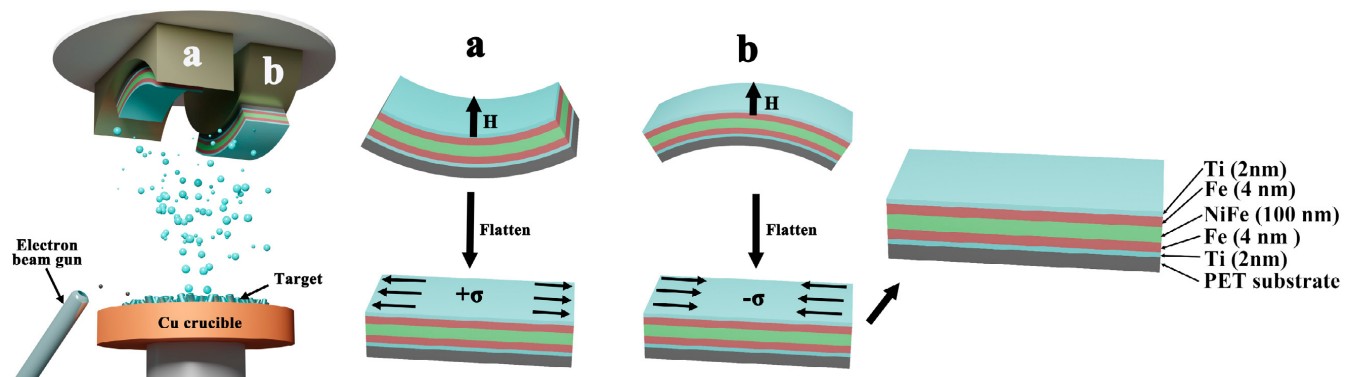

**Figure 1.** Schematic diagram of electron beam evaporation of flexible TFNFT film: (**a**) concave mold, (**b**) convex mold.

The surface morphology, surface roughness, and average grain size of the PET/TFNFT multilayer films were investigated and calculated by atomic force microscopy (AFM, Bruker MultiMode8, Bruker, Billerica, MA, USA), and its scanning area was $2 \times 2$ μm$^2$. The hysteresis loop, coercivity distribution, and magnetic film interaction of the samples were measured by using a vibrating sample magnetometer (VSM, Lake Shore 8604, Lake Shore Cryotronics, Inc., Westerville, OH, USA). Magneto-optical Kerr microscopy (MOKE, Evico Magnetics GmbH, Em-Kerr-Highres, Dresden, Germany) was used to study the domain wall displacement and magnetic domain morphology of the samples. The longitudinal Kerr effect was used to observe the magnetic domain structure. The permeability spectra were obtained using the one port short-circuit microstrip transmission line perturbation method and an Agilent network analyzer (PNA, N5227A, Keysight, Santa Rosa, CA, USA) in the frequency range of 0.1–4 GHz [27].

## 3. Results and Discussion

### 3.1. Surface Morphology Analysis

The AFM images of the TFNFT multilayer films deposited on PET substrates at various external strains are shown in Figure 2. The surface morphology's evolution reflects the growth mode change under different strains. Figure 2d shows the surface topography for the TFNFT multilayer films without external strain, which has excellent uniformity and a relatively smooth surface. When the strain is relatively small, as shown in Figure 2c,e, the grain grows vertically, improving the densification and uniformity. When the compressive strain increases, as shown in Figure 2d–g, the film's surface morphology changes from a flat needle to a valley pile shape. Due to the tensile stress and compressive stress, different surface morphologies are formed. The magnetic moment distribution in the film varies with the surface morphology. The $R_a$ and $D$ of the film are calculated from the AFM images, as shown in Figure 3. When the compressive strain is changed from $\varepsilon = 0$ to $\varepsilon = -0.78\%$, the $D$ increases from 66 nm to 88 nm. However, when the tensile strain is changed from $\varepsilon = 0$ to $\varepsilon = 0.78\%$, the $D$ increases, and large particles appear on the film surface. It is clear that the $R_a$ of the TFNFT multilayer films first decreases and then increases with strain. With the strain from $\varepsilon = -0.78\%$ to $\varepsilon = 0.78\%$, the $R_a$ of TFNFT multilayer films are 1.79 nm, 1.91 nm, 1.34 nm, 0.78 nm, 1.66 nm, 1.58 nm, and 2.41 nm. The reason may be that with increasing strain, the $R_a$ decreases slightly at the beginning of growth due to the difference in growth mode, where the film is smooth in the absence of strain. When the tensile and compressive strains increase, the microdefects increase, increasing $R_a$ and $D$. The difference in average grain size and surface morphology is caused by the restoration of the flexible substrate from the arc surface with a different radius of curvature. Therefore, the microscopic morphology of TFNFT multilayer films may change due to the diffusion mechanism or local overheating, which is not surprising for a polymer substrate, because of its low thermal conductivity [28].

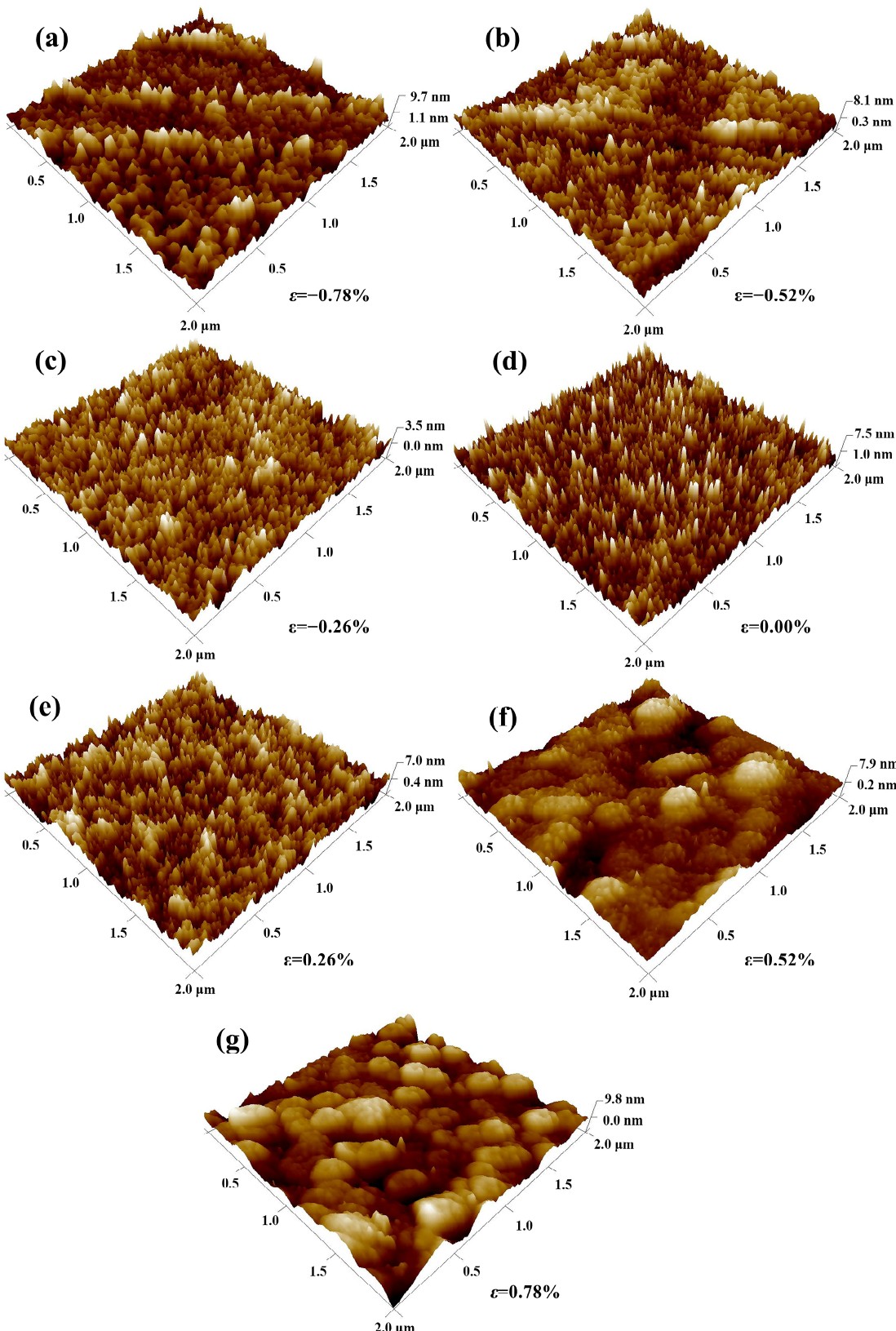

**Figure 2.** AFM images of the TFNFT multilayer films at various external strains: (**a**) $\varepsilon = -0.78\%$, (**b**) $\varepsilon = -0.52\%$, (**c**) $\varepsilon = -0.26\%$, (**d**) $\varepsilon = 0.00\%$, (**e**) $\varepsilon = 0.26\%$, (**f**) $\varepsilon = 0.52\%$, (**g**) $\varepsilon = 0.78\%$.

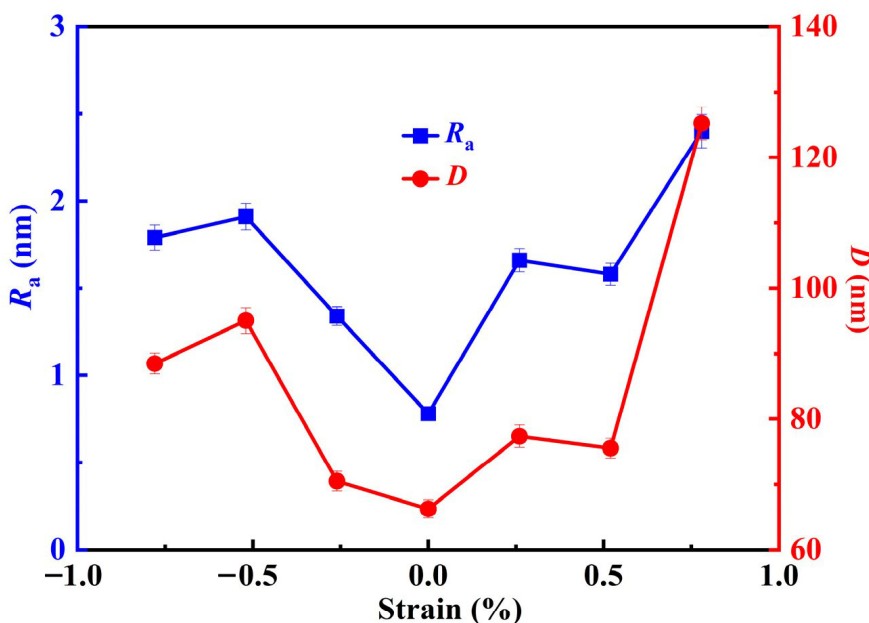

**Figure 3.** External strain dependence of surface roughness ($R_a$) and average grain size ($D$) in TFNFT multilayer films.

### 3.2. Static Magnetic Property

The magnetization ($M$)-magnetic field ($H$) hysteresis loops are measured along both the easy axis (EA) and the hard axis (HA) in the plane of the TFNFT multilayer films grown on flexible PET substrates, to investigate their magnetic properties. Figure 4 shows the in-plane magnetic hysteresis loops of the TFNFT multilayer films at various strains. In-plane measurements are mainly the characteristic of the in-plane uniaxial magnetic anisotropy. As shown in Figure 4, the magnetic hysteresis loops along the EA are rectangular in shape, while those along the HA are sheared. The coercivity ($H_c$) and the remanence ratio $M_r/M_s$ along the EA and HA obtained from Figure 4 are shown in Figure 5. The remanence ratio $M_r/M_s$ of the hysteresis loop measured along the EA increases from 0.69 to 0.92 as the tensile strain increases from $\varepsilon = 0$ to $\varepsilon = 0.78\%$, while the remanence ratio $M_r/M_s$ of the hysteresis loop measured along the EA increases from 0.69 to 0.89 as the compressive strain changes from $\varepsilon = 0$ to $\varepsilon = -0.78\%$, indicating that the magnetic moments are transversely aligned under both tensile and compressive strains. Due to the enhanced uniaxial anisotropy, the coercivity field $H_c$ measured along the EA increases from 5 Oe to 9 Oe, while the coercivity field $H_c$ for the tensile strain increases correspondingly from 6.83 to 7.38. For the hysteresis loop measured along the HA, the value of $M_r/M_s$ decreases sharply from 0.49 to 0.20 and $H_c$ increases from 7 Oe to 12 Oe with increasing compressive strain from $\varepsilon = 0\%$ to $\varepsilon = -0.78\%$. In contrast, the value of $M_r/M_s$ decreases sharply from 0.49 to 0.15 and $H_c$ increases from 7 Oe to 8 Oe with increasing tensile strain from $\varepsilon = 0\%$ to $\varepsilon = 0.78\%$. It follows that the EA of the film can be tuned to the HA by applying tensile strain in the hard magnetization direction or compressive strain in the easy magnetization direction of the film. Conversely, the HA of the film can be tuned to the EA. Therefore, the magnetic anisotropy of the film can be effectively regulated by strain. The distinctive features of uniaxial magnetic anisotropy are most likely generated by residual stresses caused by the deformation of the PET substrate. The strain-induced uniaxial magnetic anisotropy $H_K$ is closely related to the field corresponding to the intersection of the HA and EA hysteresis loops [29]. It is obvious from Figure 4 that $H_K$ increases gradually with increasing tensile strain, while $H_K$ decreases gradually with increasing compressive strain. The stress has an obvious regulatory effect on the rectangular ratio and the corrective force of the magnetization curve. In the direction of the easy magnetization axis of the TFNFT film, the correction force increases with the enhancement of the stress-induced anisotropy.

This is mainly because the enhancement of the magnetic anisotropy will hinder the rotation of the magnetic domain, thus forming hysteresis, which increases the correction force of the TFNFT film. However, when the magnetic field is perpendicular to the easily magnetized axis, the control of compressive stress on the coercivity is not obvious.

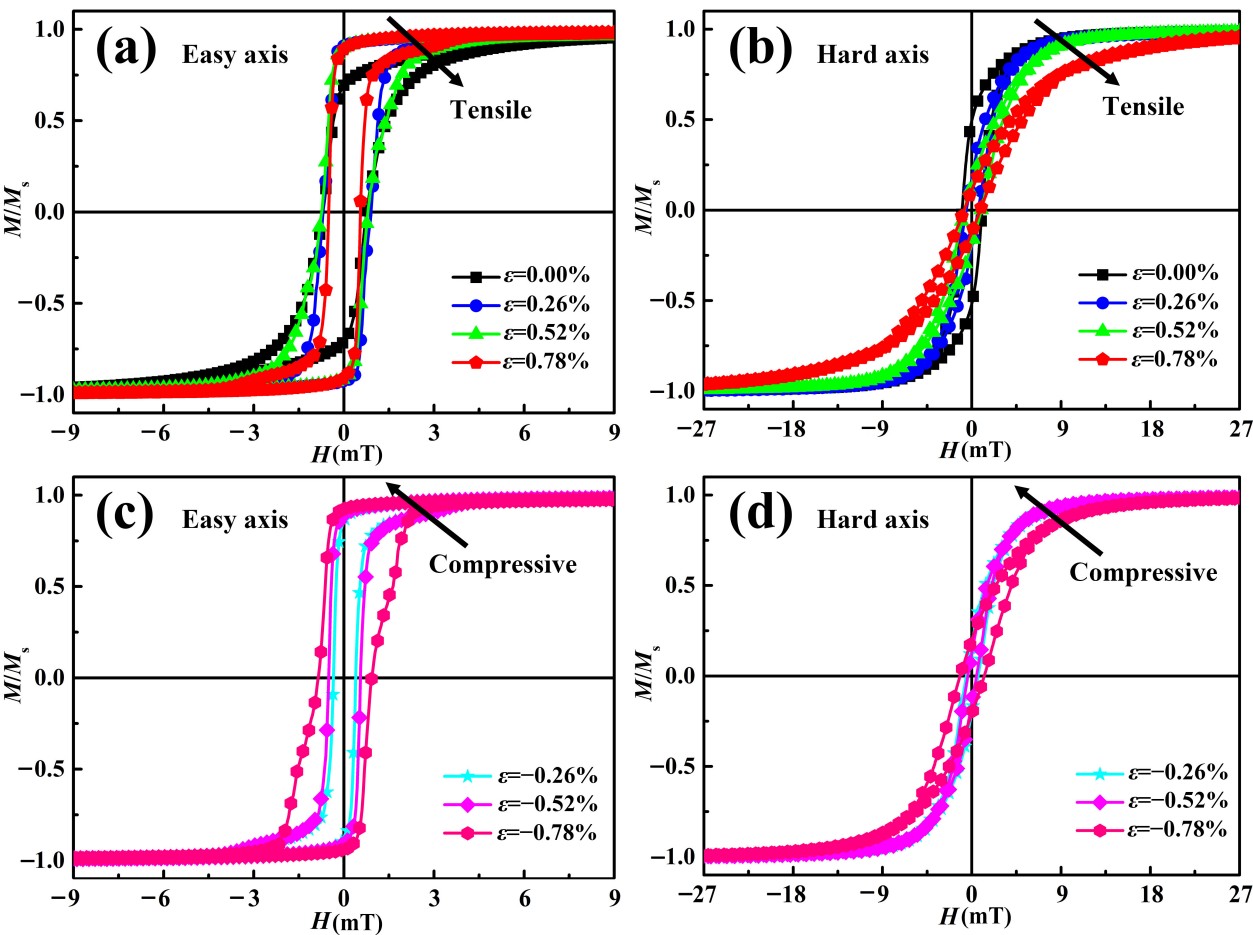

**Figure 4.** Magnetic hysteresis loops of TFNFT multilayer films under various external strains: (**a**) tensile strain along the easy axis (EA), (**b**) tensile strain along the hard axis (HA), (**c**) compressive strain along the EA, (**d**) compressive strain along the HA.

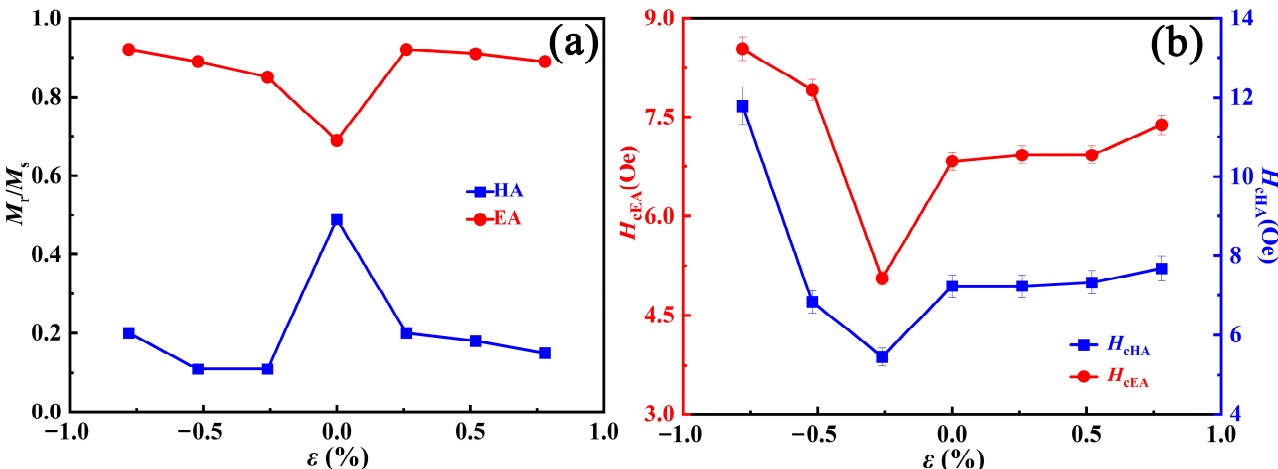

**Figure 5.** Static magnetic properties of TFNFT multilayer films under various external strains: (**a**) remanence ratio $M_r/M_s$ in the EA and HA, respectively; (**b**) the coercivity ($H_c$).

Figure 6 shows the FORC diagrams of TFNFT multilayer films at various external strains. In Figure 6, the coercivity field distributions exhibit closed profiles, indicating that the samples are in a single-domain state [30,31]. The coercivity distribution of the tensile and the compressive strain in the HA and the EA are distributed wider in the transverse axis direction, compared to the coercivity distribution in the absence of external strain, due to greater coercivity. The vertical axis of the 2-D FORC diagram shows different magnetic interaction, with the strain being $\varepsilon = -0.78\%$, $\varepsilon = 0$, and $\varepsilon = 0.78\%$. A broader distribution along the $H_u$ axis can be seen in Figure 6b,e, indicating the existence of strong magnetic interaction. From Figure 6d, it can be seen that the coercivity distribution is relatively wide, which is due to the appearance of aligned peaks in the surface morphology (as shown in Figure 2f). It affects the distribution of coercivity and magnetic interactions when the surface micromorphology clusters into packets (as shown in Figure 2a). Therefore, the changes in surface micromorphology are closely related to the distribution of coercivity.

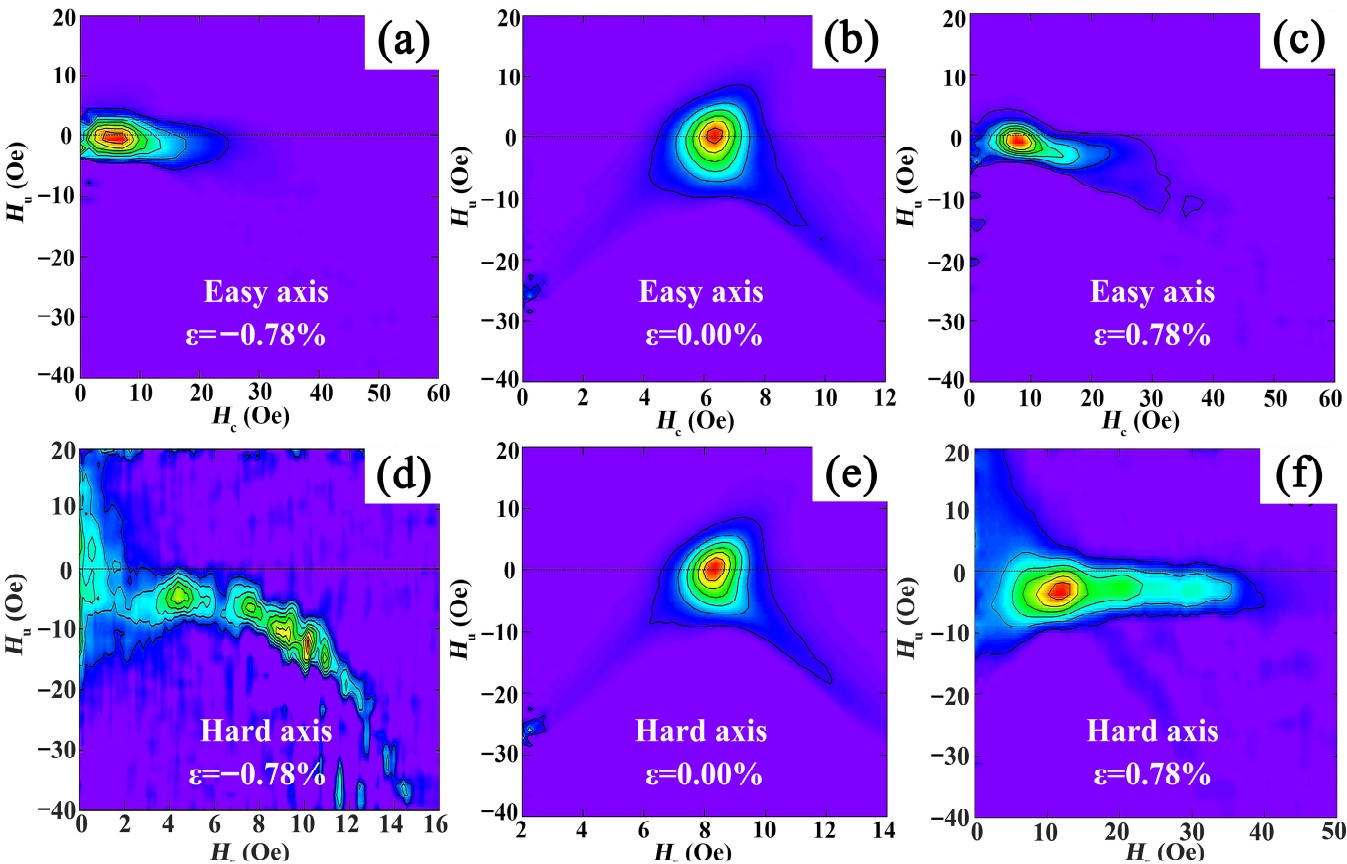

**Figure 6.** FORC diagrams of TFNFT multilayer films at various external strains: (**a**) $\varepsilon = -0.78\%$, (**b**) no strain, (**c**) $\varepsilon = 0.78\%$ (along the EA); (**d**) $\varepsilon = -0.78\%$, (**e**) no strain, (**f**) $\varepsilon = 0.78\%$ (along the HA).

### 3.3. Magnetic Domain Analysis

MOKE measurements of the TFNFT multilayer films are performed, and the images are shown in Figure 7. The dark and light colors represent the two different states of magnetization with the magnetic moment perpendicular to the film face downward and upward, respectively. The MOKE images in the sample when the compression strain is $\varepsilon = -0.78\%$ is shown in Figure 7($a_1$–$f_1$). When the external magnetic field is applied, all magnetic moments appear upward (as shown in Figure 7($a_1$)). In Figure 7($b_1$), the external magnetic field gradually increases to $-1.10$ mT, and the domain wall is displaced. When the external magnetic field is increased to $-1.59$ mT, the magnetic moment is completely reversed in the first step, as shown in Figure 7($d_1$). The external magnetic field c reverses and increases to 1.21 mT, and the domain wall shifts, as shown in Figure 7($e_1$). When

the external magnetic field increases to 1.51 mT, the magnetization state of the magnetic moment becomes completely perpendicular to the film surface (Figure 7(f$_1$)). The whole process of flipping the magnetic domains in the sample is achieved by changing the magnitude of the external magnetic field perpendicular to the film surface.

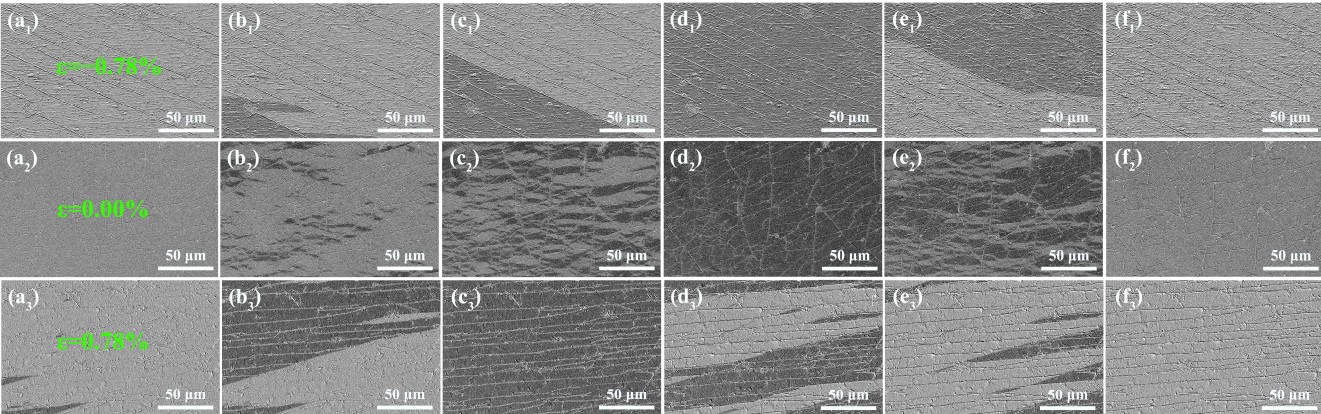

**Figure 7.** MOKE images of Ti/Fe/Ni$_{81}$Fe$_{19}$/Fe/Ti multilayer films. ((**a$_1$**)–(**f$_1$**)) indicate variations in the magnetic domain under an external strain of $\varepsilon = -0.78\%$: (**a$_1$**) −0.94 mT, (**b$_1$**) −1.10 mT, (**c$_1$**) −1.20 mT, (**d$_1$**) −1.59 mT, (**e$_1$**) 1.21 mT, and (**f$_1$**) 1.51 mT; ((**a$_2$**)–(**f$_2$**)) indicate variations in the magnetic domain under no external strain: (**a$_2$**) −1.42 mT, (**b$_2$**) −2.58 mT, (**c$_2$**) −2.72 mT, (**d$_2$**) −4.01 mT, (**e$_2$**) 2.66 mT, and (**f$_2$**) 4.03 mT; ((**a$_3$**)–(**f$_3$**)) indicate variations in the magnetic domain under an external strain of $\varepsilon = 0.78\%$: (**a$_3$**) −0.68 mT, (**b$_3$**) −0.88 mT, (**c$_3$**) −1.57 mT, (**d$_3$**) −0.82 mT, (**e$_3$**) 0.93 mT, and (**f$_3$**) 1.76 mT.

The MOKE images of the magnetic domain change in the sample when there is no external strain, as shown in Figure 7(a$_2$–f$_2$). As shown in Figure 7(a$_2$–d$_2$), when the external magnetic field is −1.42 mT, the magnetization state of the magnetic moments in the film surface is perpendicular to the film surface. Gradually increasing the external magnetic field to −2.58 mT, the domain wall nucleation appears. As the external magnetic field increases to −4.01 mT, the growth, expansion, and overturning process occurs sequentially. The MOKE images of the magnetic domain change in the sample when the tensile strain is $\varepsilon = 0.78\%$, as shown in Figure 7(a$_3$–f$_3$). When the sample is subjected to tensile and compressive strain, the magnetic moment is reversed consistently over a large range. It shows a dendritic expansion process, which finally achieves the reversal process of the magnetic domain in the sample when the sample is not subject to external strain. It can be seen that the strain has a strong influence on the magnetic domains of the film. As the strain increases, the influence of the degenerate magnetic field in the sample decreases and the in-plane anisotropy increases.

### 3.4. High-Frequency Properties

Based on the Kittle equation [26], the $f_r$ of magnetic films depends on the in-plane uniaxial anisotropy field. According to Figure 4, the in-plane anisotropy field increases with the strain. The complex permeability spectra of TFNFT multilayer films grown with various pre-strains are shown in Figure 8, to evaluate the impact of strain on the dynamic characteristics of TFNFT multilayer films. The LLG (Landau–Liftshitz–Gilbert) equation is used to fit the experimental results [32]. It is clear from Figure 8 that the experimental data is consistent with the fitted curve. With the compressive strain increasing from $\varepsilon = 0\%$ to $\varepsilon = -0.78\%$, $\mu_i$ decreases from 385 to 80, while $f_r$ increases from 1.9 to 3.4 GHz, as shown in Figure 8a–d. The result reflects the positive correlation between the in-plane uniaxial magnetic anisotropy and the $f_r$ of the TFNFT multilayer films. The initial susceptibility decreases with increasing strain. The opposite features can be explained by Snoek–Archer's limit and are associated with an exchange behavior between $f_r$ and permeability [32,33].

With the tensile strain increasing from $\varepsilon = 0\%$ to $\varepsilon = 0.78\%$, $\mu_i$ decreases from 385 to 300, while $f_r$ increases from 1.9 to 2.5 GHz, as shown in Figure 8d–g. For all samples, as the frequency is increased, there is the displacement and split of the peaks. At the higher frequencies, the Villary effect is mainly due to the resonance frequency effect [34]. It can be seen that the compressive strain can effectively modulate the $f_r$ compared to the tensile strain.

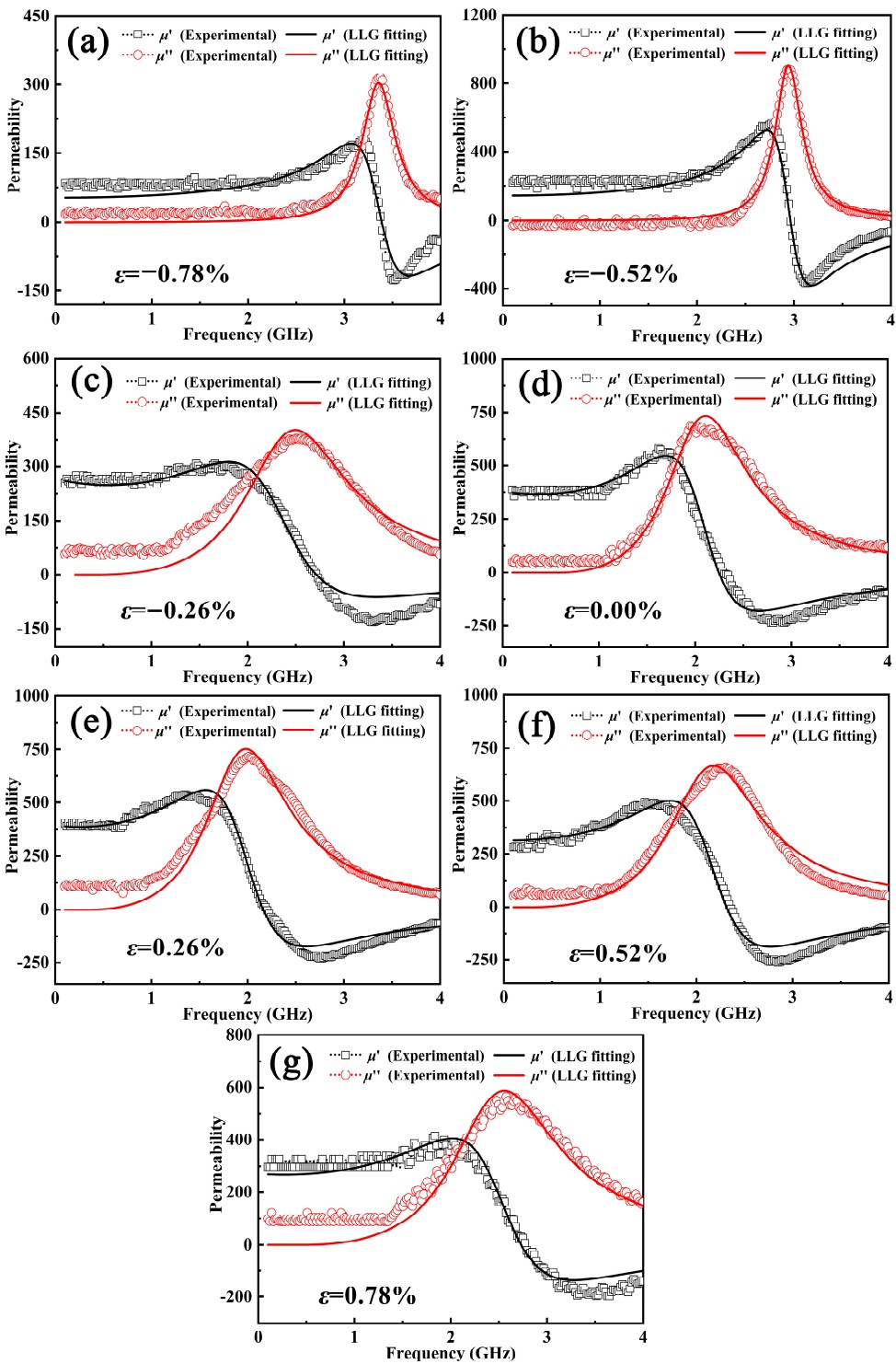

**Figure 8.** Permeability spectra of TFNFT films at various external strains: (**a**) $\varepsilon = -0.78\%$, (**b**) $\varepsilon = -0.52\%$, (**c**) $\varepsilon = -0.26\%$, (**d**) no strain, (**e**) $\varepsilon = 0.26\%$, (**f**) $\varepsilon = 0.52\%$, (**g**) $\varepsilon = 0.78\%$.

## 4. Conclusions

In summary, we employed a pre-stretched and compressed surface structure on PET to fabricate flexible TFNFT multilayer films. We reported the effects of the strain on the microscopic morphology, magnetic domain morphology, and magnetic properties of TFNFT multilayer films. They showed uniaxial magnetic anisotropy and displayed good high-frequency performance, with $\mu_i$ varying from 80 to 385 and $f_r$ varying from 1.9 to 3.4 GHz by varying the pre-strain. When pre-strained uniaxial tensile and compressive strains were applied to the TFNFT film, the surface morphology of the sample exhibited different states. Moreover, $\mu_i$ c varied continuously around a few hundred units, and $f_r$ varied in the range of 1.5 GHz. These results demonstrate a simple and convenient method for continuously tuning the high-frequency characteristics of flexible magnetic films by applying uniaxial strain. Flexible magnetic films have a wide range of tunable resonance frequencies, which paves a new way to develop flexible high-frequency devices with adjustable mechanical strain. Moreover, it plays a vital role in magnetic-integrated device applications.

**Author Contributions:** Conceptualization, Z.H., X.J. and K.S.; Methodology, Z.H., Z.M., Y.D., J.Y., X.J. and K.S.; Writing-original draft, Z.H.; Writing—review & editing, Z.H., Q.L., Z.L. (Ziyu Li) and K.S.; Investigation, C.W. (Chuanjian Wu); Software, C.W. (Chaoming Wang); Visualization, Z.Y. and Z.L. (Zhongwen Lan); Project administration: Z.Y.; Validation, Z.L. (Zhongwen Lan); Funding acquisition, K.S.; Supervision, K.S. All authors have read and agreed to the published version of the manuscript.

**Funding:** This present work was financially supported by the National Natural Science Foundation of China under Grant Nos. 52172267 and 51772046, and the Innovation Group Project of Sichuan Province (2022JDTD0018).

**Institutional Review Board Statement:** Not applicable.

**Informed Consent Statement:** Not applicable.

**Data Availability Statement:** Data underlying the results presented in this paper are not publicly available at this time but may be obtained from the authors upon reasonable request.

**Conflicts of Interest:** The authors declare no conflict of interest.

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
