# Peer review of "Strain Modulation of Microstructure, Magnetic Domains, and Magnetic Properties of Ti/Fe/Ni81Fe19/Fe/Ti Multilayer Thin Films"

_coatings, doi:10.3390/coatings13020363_

Round 1
Reviewer 1 Report
Interesting topic and well, but not described in depth and scientifically exact.
However, it is questionable whether only surface images, subchapter 3.1 line 116, can be used to describe the microstructure. Better to use surface morphology here. Similarly, the mathematical description variables such as RMS, roughness depth, bearing ratio should be given of all surface images. Sometimes strange values come out here, but in the overall context they might explain the layer behaviour.
Oerstedt (Oe) is not an SI unit for magnetic field strength. It has not been considered an official unit since 1970. Convert to Tesla or µT here!
Some changes in style are also necessary!
A space belongs after a number and unit of measurement, but a protected space so that no line break happens, e.g. line 104
The labelling of the drawing files of figure (a) - (g) is done with a font that is too large.
Author Response
Please see the attachment。

Reviewer 3 Report
Submitted manuscript entitled “Strain modulation of microstructure, magnetic domains and magnetic properties of Ti/Fe/Ni81Fe19/Fe/Ti multilayer thin films” is devoted to the design and testing a simple and convenient method by continuously applying uniaxial tensile strains to tune the high-frequency properties of flexible magnetic films. Despite the fact that approach is not exactly new and it was previously used (Garcia et al. Induced anisotropy, magnetic domain structure and magnetoimpedance effect in CoFeB amorphous thin film. J. Magn. Magn. Mater. 191 (1999) 339; Correa et al. Exploring the magnetization dynamics, damping and anisotropy in engineered CoFeB/(Ag, Pt) multilayer films grown onto amorphous substrate. J. Magn. Magn. Mater. 485 (2019) 75) the subject is good for this journal and work contains some interesting results.
The authors must explain the reasons for the appearance of magnetic anisotropy in the film plane, which is important for the results understanding. In addition, it is necessary to indicate the orientation of the easy magnetization axis with respect to the sides of the sample and add the appropriate notation in Fig. 1.
It follows from Fig. 4 that, when measuring the hysteresis loops, the applied magnetic field change interval was at least 1 Oe. With such a step of changing the field, it is impossible to determine the value of Hc with an accuracy of 0.01 Oe, as the authors claim (see, for example, page 7). With this respect it is also important to add error bars for the data reported in the Fig. 5.
The authors should indicate which configuration of the Kerr effect was used in the observation of the magnetic domain structure.
Why the data presented in the contradictory way: Fig. 4 shows in-plane hysteresis loops, but when observing the domain structure, the field was applied in a direction perpendicular to the film surface?
The grain size defined by the microscopy can be defined with big error. Usually, it is done using X-ray analysis or microscopy. Authors discuss 0.78 nm roughness value in the absence of strain. For this type of the films careful comparison of the grain size and properties, including high frequency dynamics, for the case of the flexible substrates was reported in the literature (Agra et al., J. Magn. Magn. Mater. 355 (2014) 136; Kurlyandskaya et al. J. Magn. Magn. Mater. 415 (2016) 91). Careful discussion of the problem must be added.
When measuring hysteresis loops and the changes of the magnetic domains, the authors use units from different measurement systems, as units of magnetic field measurement only one system is allowed.
Page 3, 113-115 lines: Unnecessary service phrase.
Page 4, line 123: “Figure 2 from (d) to (a)” should be “Figure 2 from (d) to (g)”.
The manuscript requires a major revision.
Round 2
Reviewer 1 Report
My comments on the unit of magnetic field strength Oerstedt and the further use, although Tesla has stood for it since 1970, were not heeded.
Reviewer 2 Report
The authors appropriately responded to the comments and accordingly revised the manuscript.
Author Response
Thanks very much for reviewer’s comments.
Reviewer 3 Report
Authors made some improvements in the text of the submitted revision but they left one of the important questions badly answered. There were many studies of FeNi based materials deposited onto different substrates including flexible materials. For this type of the films careful comparison of the grain size and properties, including high frequency dynamics, for the case of the flexible substrates was reported in the literature and careful discussion of the problem must be added (Agra et al., J. Magn. Magn. Mater. 355 (2014) 136; Kurlyandskaya et al. J. Magn. Magn. Mater. 415 (2016) 91; etc.). It is especially important in a view of poor referencing offered for audience by the authors mostly focused on very few groups working in the field.
